# An Evaluation of Serological Tests to Determine Postvaccinal Immunity to SARS-CoV-2 by mRNA Vaccines

**DOI:** 10.3390/jcm11247534

**Published:** 2022-12-19

**Authors:** Graciela Iglesias García, Ángel Díaz Rodríguez, Beatriz Díaz Fernández, Carmela Cuello Estrada, Tania García Ferreiro, Noelia Crespo García, Jesús Seco-Calvo

**Affiliations:** 1Institute of Biomedicine (IBIOMED), Campus of Vegazana, University of Leon, 24071 Leon, Spain; 2Bembibre Health Center, University of Leon, Carbajal Street 1, 24300 Bembibre, Spain; 3Arturo Eyries Center, Puerto Rico Street 1, 47014 Valladolid, Spain; 4Physiotherapy Department, Institute of Biomedicine (IBIOMED), Campus of Vegazana, University of Leon, 24071 Leon, Spain; 5Psychology Department, Faculty of Medicine, Basque Country University, 48900 Leioa, Spain

**Keywords:** automated immunoassays, COVID-19, lateral flow immunoassays, performance, SARS-CoV-2, ELISA immunoassays, CLIA immunoassays

## Abstract

Background: The duration of the protective efficacy of vaccines against SARS-CoV-2 is unknown. Thus, an evaluation of the clinical performance of available tests is required. Objectives: To evaluate the clinical performance of LFIA immunoassay compared to ELIA and CLIA immunoassays available in Europe for the detection of IgG antibodies generated by mRNA vaccines against SARS-CoV-2. Methods: Two automated immunoassays (the EUROIMMUN anti-SARS-CoV-2 IgG S1 ELISA and the LIAISON de Diasorin anti-SARS-CoV-2 IgG S1/S2 test) and a lateral flow immunoassay (the Livzon LFIA anti-SARS-CoV-2 IgG S test) were tested. We analyzed 300 samples distributed in three groups: 100 subjects aged over 18 years and under 45 years, 100 subjects aged between 45 and 65 years, and 100 subjects aged over 65 years. The samples were collected before vaccination; at 21 days; and then at 1, 2, 3, and 6 months after vaccination. The sensitivity, specificity, positive predictive value, negative predictive value, positive probability quotient, negative probability quotient, and concordance (kappa index) were calculated for each serological test. Results: The maximum sensitivity values for IgG were 98.7%, 98.1%, and 97.8% for the EUROIMMUN ELISA, Abbott CLIA, and Livzon LFIA tests, respectively, and the maximum specificity values for IgG were 99.4%, 99.9%%, and 98.4% for the ELISA, CLIA, and LFIA tests, respectively, at the third month after vaccination, representing a decrease in the antibody levels after the sixth month. The best agreement was observed between the ELISA and CLIA tests at 100% (k = 1.00). The agreement between the ELIA, CLIA, and LFIA tests was 99% (k = 0.964) at the second and third month after vaccination. Seroconversion was faster and more durable in the younger age groups. Conclusion: Our study examined the equivalent and homogeneous clinical performance for IgG of three immunoassays after vaccination and found LFIA to be the most cost-effective, reliable, and accurate for routine use in population seroconversion and seroprevalence studies.

## 1. Introduction

Vaccination against SARS-CoV-2 has proven to be the most effective measure to control the COVID-19 pandemic. COVID-19 vaccines have been shown to be effective in preventing infection, serious illness, hospitalization, and death [1].

Naturally acquired immunity confers stronger protection against infection and symptomatic disease caused by SARS-CoV-2 compared to the two-dose immunity given by mRNA vaccines [2]. Natural infection can produce B-cell responses that continue to evolve for at least a year. The results of one study suggest that an increase in people vaccinated with currently available mRNA vaccines will increase the neutralizing activity of plasma but may not produce antibodies with an amplitude equivalent to that obtained by vaccination of convalescent individuals [3].

Clinical trials have shown that mRNA vaccines (Pfizer-BioNTech BNT162b2 and Moderna’s mRNA-1273 vaccines) provide strong protective efficacy against COVID-19 and are highly effective in the first months after vaccination against documented infection and symptomatic COVID-19 [4,5,6,7,8,9]. These mRNA vaccines induce a lasting immune memory to SARS-CoV-2 and its worrisome variants at least 6 months after vaccination [10].

However, there have been studies focusing on the protective duration of these vaccines and the decreasing trend of antibody levels. Several studies have indicated that immunity gradually decreases in all age groups a few months after receiving the second dose of the vaccine [11,12,13,14]. As a result, the confirmed infection rate among vaccinated people significantly increases as the time since vaccination increases [11,12,13,14], reinforcing the necessity for booster doses to combat COVID-19.

The decrease in antibody levels and the increase in infections months after vaccination point to decreased immunity from vaccination with time [15,16,17]. However, despite the decrease in antibody levels, protection against serious diseases and hospitalization remains high [18], which suggests that persistent cellular immunity boosts the immune response and prevents viral spread when antibodies disappear [19].

The rapid and accurate measurement of SARS-CoV2-specific neutralizing antibodies is essential to monitor immunity in infected and vaccinated subjects. The current gold standard is based on the microneutralization (MNA) test, which requires the use of specialized biosecurity containment facilities and the use of live pathogens [20,21], and the pseudovirus neutralization test (pVNT), which requires research laboratories, indicating that both tests are not suitable for large populations. Consequently, fast and reliable high-performance neutralization tests are needed to detect neutralizing antibodies in different environments and populations.

Vaccines cause high levels of antireceptor binding domain (RBD) [22] antibodies and/or are directed against the protein S of the spike [23]. Antibodies directed against the S1 subunit of the protein S of SARS-CoV-2, specifically against RBD, have been shown to strongly correlate with the neutralization of the virus [24]. In order to correctly interpret the results of serological tests, the type of test used and the neutralizing antibodies generated by the vaccine and natural infection should be taken into account. Several studies have evaluated the correlation between serological tests that detect binding antibodies (RBDs) with neutralizing activity to provide information on the functional capabilities of the detected antibodies [25,26,27].

Different assays have been marketed. For example, automated assays (enzyme immunoabsorption assays (ELISAs) or immunoassays of chemiluminescent enzymes (CLIAs)) or rapid screening assays (lateral flow immunoassays (LFIAs)) can be a very useful tools to identify people who have antibody neutralizers against SARS-CoV-2. The ALS, CLIA, and LFIA tests seem to be a very useful in identifying people who have neutral antibodies against SARS-CoV-2 and are potentially immune; performing epidemiological studies; and evaluating the effectiveness of vaccines, the seroconversion rates, and the duration of humoral immunity produced by vaccines [28,29,30,31,32].

In this study, we evaluated the clinical performance of two quantitative immunoassays, namely, ELIA and the CLIA, and compared them with the performance of an LFIA immunoassay that qualitatively measures IgG antibodies against protein S, which has not been validated in any study as a method of evaluating and monitoring postvaccine humoral reduction. The LFIA immunoassay allows us to understand the presence of IgG by easily, cheaply, and quickly offering results in less than 15 min. In addition, it allows us to identify persons who are potentially immune with neutralizing antibodies against SARS-CoV-2.

The evaluation was carried out on healthy subjects vaccinated with mRNA vaccines by comparing sensitivity values, specificity values, predictive values, the likelihood ratio, the percentage of agreement, and the degree of Cohen’s kappa agreement, with a follow-up of 6 months, to determine the evolution in the performance of immunoassays over time and their variability.

In this study, we intended to answer the following research question: Are LFIA immunoassays useful compared to quantitative immunoassays for the detection of humoral immunity (IgG) provided by mRNA vaccines?

This study aimed to evaluate the clinical performance of LFIA immunoassay compared to ELIA and CLIA immunoassays available in Europe for the detection of IgG antibodies generated by mRNA vaccines against SARS-CoV-2.

## 2. Materials and Methods

### 2.1. Design and Participants

This prospective observational study included 300 healthy subjects without previous infection who were vaccinated with two doses of mRNA vaccines (Comirnaty from Pfizer/BioNTech at 1–21 days and the Moderna mRNA vaccine at 1–28 days) and were tested for IgG antibodies for 6 months after vaccination. The study was conducted in the province of León in collaboration with the University of León in 2021.

The samples were collected directly from patients selected and included in the study at vaccination points before vaccination by performing RT-PCR to exclude infection. The IgG was measured by the ELISA test to determine their immune status. After vaccination, the samples were collected at 21 days and then at 1, 2, 3, and 6 months.

Subjects were recruited through systematic sampling with random start from the SACYL database of 115,075 candidates to receive mRNA vaccines (9.6% with Moderna) in the province of León until 300 subjects were selected at the time of vaccination. They were then distributed into 3 age groups—100 subjects aged over 18 years and under 45 years; 100 subjects aged between 45 and 65 years; and 100 subjects aged over 65 years—to avoid possible age-induced variability. Healthy vaccinated people were included instead of people with a history of SARS-CoV-2 infection in order to compare more or less standardized immunoassays. All the participants gave informed consent to participate in the full vaccination dose study. Another inclusion criterion was age of >18 years, excluding those who did not sign the informed consent, had incomplete vaccination pattern, had active infection, had positive GI antibodies, or did not want to participate in the study. The study was carried out in accordance with the Declaration of Helsinki and was approved by the Institutional Review Board (or the Ethics Committee) of the León Hospital (Spain) (protocol code 2164 on 30/03/2021”) for studies involving humans.

### 2.2. Serological Tests

Samples of subjects vaccinated against COVID-19 with positive and negative IgG antibodies against the S1/S2 tests of the receptor-binding domain (RDB) were determined with two automated immunoassays (the EUROIMMUN anti-SARS-CoV-2 IgG ELISA against the S1 domain of the herringbone protein and LIAISON SARS-CoV-2 S1/S2 and DiaSorin CLIA IgG assays) as gold-standard reference tests and a lateral flow immunoassay (Livzon diagnostic kit for IgG antibody against SARS-CoV-2) as a test used against protein S.

Clinical evaluation of the tests was performed by calculating the sensitivity, specificity, positive predictive value (PVP), negative predictive value (PVN), positive probability ratio (CPP), and negative probability ratio (CPN) as well as the correlation using Cohen’s kappa index for each serological test. If the degree of concordance meets the above clinical requirements, both methods would be considered equivalent. To verify the accuracy and applicability of the test in clinical practice, the LFIA immunoassay must have a sensitivity, specificity, and degree of concordance greater than 90%.

#### 2.2.1. The ELISA Test

The EUROIMMUN anti-SARS-CoV-2 IgG S1 ELISA (EUROIMMUN, Lüebeck, Germany) test was carried out according to the manufacturer’s guidelines on the DS2 system, an automated microplate technology (Dynex Technologies GmbH, Den-kendorf, Germany). The microplate wells were coated with recombinant structural protein S1, and the assay specifically detected IgG antibodies against SARS-CoV-2 using the S1 domain of the spike protein, including the immunologically relevant receptor-binding domain (RBD). This is a very useful technique for the quantitative detection of IgG anti-S1 humoral immune response to perform studies of seroprevalence or postvaccine seroconversion. It has excellent performance and good correlation with different test systems for the detection of neutralizing antibodies confirmed in other immunoassays. The results were evaluated as follows: <0.8—negative; 0.8 to <1.0—limit; and 1.1—positive [33].

#### 2.2.2. The CLIA Test

The LIAISON test of Diasorin anti-SARS-CoV-2 IgG S1/S2, conceived and developed in Gerenzano (Italy), guarantees extremely precise results with a sensitivity of 97.4% and a specificity of 98.5%. The test is based on chemiluminescence technology (LCIS) for quantitative and qualitative analysis as well as IgG antibody determination against SARS-CoV-2 S1 and S2 proteins in serum or human plasma samples without cross-reactions with other circulating human coronaviruses. This test identifies neutralizing antibodies and therefore represents an important tool to study the immune response against SARS-CoV-2 vaccines. The quantification range was between 3.8 and 400.0 AU/mL. The limit for positivity was >15 AU/mL, so results between 12.0 and 15.0 AU/mL were considered as reaching their limit [34].

#### 2.2.3. The Lateral Flow Test (LFIA)

The Livzon anti-SARS-CoV-2 IgG S test is a qualitative immunocolloidal lateral flow immunoassay (LFIA) that detects IgG antibodies directed at the RBD domain and against the S protein of the ear to identify people with neutralizing antibodies against SARSCoV-2 in serum or plasma. It is an immunocolloidal test for the qualitative detection of IgG antibodies against protein S. In this study, 10 µL was added. The results were read and interpreted 15 min after the test as positive or negative depending on whether the antibody band was colored or not.

### 2.3. Statistical Analysis

All the statistical analyses were performed using IBM SPSS 21.0 Statistics (Statistical Package for Social Sciences, IBM Corp., Chicago, IL, USA) and Microsoft Excel 2016 software. To evaluate the sensitivity, specificity, positive and negative predictive values, and positive and negative probability coefficients, we chose the ELISA and CLIA assays as gold standard. Positive agreement percentage and Cohen’s kappa in all samples were collected before vaccination; at 21 days; and at 1, 2, 3, and 6 months after vaccination. With a range between 0 and 1, a kappa value of 0.40 denotes poor agreement, a value between 0.40 and 0.75 denotes fair/good agreement, and a value above 0.75 denotes excellent agreement. A value of *p* < 0.05 was considered statistically significant, and a 95% confidence interval (CI) was reported for each metric. The sensitivity, specificity, positive predictive value (PVP), negative predictive value (PVN), positive probability ratio (CPP), and negative probability ratio (CPN) were calculated for each serological test.

To verify the accuracy and applicability of the test in clinical practice, it must have a degree of sensitivity, specificity, general agreement, and a degree of concordance greater than 90%. Finally, the percentages of postvaccination seroconversion were compared with the three immunological trials by age group.

## 3. Results

The subjects of the sample (N = 300) had an average age of 58.12 years, of which 62.2% were men, d there were no significant differences by age and sex. By age group, the average was 28.9 years in the 100 subjects aged over 18 years and under 45 years, 56.8 years in the 100 subjects aged between 45 and 65 years, and 83.9 years in the 100 subjects aged over 65 years. All were healthy at the time of vaccination with a negative PCR-RT, and negative IgG antibodies were determined by the ELISA test.

Table 1 summarizes the sensitivities, specificities, positive and negative predictive values, and probability coefficients.

### 3.1. Sensitivity

The IgG ELISA sensitivity was 60.3% after the first dose and increased to 95.8% at 2 months after vaccination. It reached its maximum at 3 months with 98.7% and decreased to 83.1% after 6 months. The sensitivity for the CLIA was very similar, increasing from 60.4% to 86.1%, 96.2%, and 98.1% at the third month before decreasing to 80.3% at the sixth month. The sensitivity of LFIA behaved similarly, increasing from 60.6% to 80.2%, 94.8%, and 97.8% until the third month before decreasing to 83.1% from the sixth month. The general sensitivity for postvaccine IgG was equivalent (95%) for the ELISA, CLIA, and LFIA tests and reached its maximum between the second and third postvaccination months before then declining from the sixth month, thus coinciding with a decrease in the level of protective antibodies. A comparison of susceptibility during the first 21 days of vaccination did not reveal any significant differences between the three assays, being significant from one month (*p* < 0.05).

### 3.2. Specificity

The general specificity values were equivalent for the ELISA, CLIA, and LFIA tests (higher than 98) and reached their maximum between the second and third postvaccination months before then declining after the sixth month, thus coinciding with a decrease in the level of protective antibodies. The specificity was significantly different between the three trials from the beginning (*p* < 0.05). In addition, overall, the three trials had very homogeneous values, i.e., higher than 90% in the positive and negative predictive values and reaching the maximum between the first and the third months.

### 3.3. The Likelihood Ratio

The positive likelihood ratio steadily increased between 21 days and 3 months and was higher between 2 and 3 months for CLIA and the LFIA than for ELISA.

### 3.4. The Concordance of IgG Serologic Tests

Table 2 summarizes the general agreement and agreement regarding the timing of IgG determinations from the start of vaccination to 6 months. Overall, an excellent agreement was observed between the three trials, even from day 21 of vaccination. The greatest concordance was observed between the ELISA and CLIA tests at 100% (k = 1.00) in the second and third months after vaccination.

### 3.5. The Immune Response

The immune response generated by mRNA vaccines was determined with the three immunoassays in the different age groups. As shown in Figure 1, younger patients showed a faster and more durable response than patients aged over 65 years of age throughout the follow-up period.

### 3.6. Seroconversion Panel

We reliably compared our Livzon LFIA test with the gold-standard microneutralization test on a seroconversion panel. This panel consisted of eight plasma samples from a single donor vaccinated with IgG neutralizing antibodies performed by the pseudovirus neutralization test distributed by Access Biologicals (CVD19SCP), and the degree of negative and positive concordance of the test against two commercial kits for the detection of antibodies in people vaccinated against SARS-CoV-2 was also determined.

As shown in the Table 3, the results obtained with the Livzon test tenuously provided limit values (−/+) at 21 days after vaccination and were very positive in samples collected per month of vaccination, similar to other commercial kits, offering a positive degree of concordance between 99% and 100% and a negative degree of concordance between 88% and 100%.

## 4. Discussion

In general, antibodies are considered the most accurate method available for determining protection against COVID-19 as they correlate with the immune response’s ability to neutralize the virus’ entry into human cells [28]. Clinical testing for IgG antibodies produced by vaccines is essential to understand the immune status of the population and the duration of immunity. In general, the number of samples presenting IgG antibodies against SARS-CoV-2 is related to the number of doses and days since the start of vaccination but differs with the type of test.

In this work, we evaluated the clinical performance of two quantitative-type immunoassays, namely, ELIA and CLIA, compared to the performance of LFIA with CE marking for the detection of IgG antibodies SARS-CoV-2 in human serum and plasma in subjects vaccinated against COVID-19 with two doses of mRNA vaccines.

In our study, we showed maximum IgG sensitivity values of 98.7%, 98.1%, and 97.8% for the EUROIMMUN ELISA, LIAISON CLIA, and Livzon LFIA tests, respectively, at 3 months after vaccination. We also showed maximum IgG specificity values of 99.4%, 99.9%, and 98.4% for the ELISA, CLIA, and LFIA tests, respectively, at 3 months after vaccination with a decrease in antibody levels from the sixth month.

The sensitivity to IgG ranged between 94% and 100% after natural infection, and the specificity values were between 99% and 100%. The LFIA test had 95% sensitivity and 97% specificity for diagnosis.

The performance of the EUROIMMUN ELISA trial has been evaluated in some studies, showing sensitivity values for IgG between 85% and 95% after natural infection and specificity values between 95% and 100% [29,30,32,33,35]. In a prospective study conducted in health workers to monitor the adaptive immune response in the medium (3 months) and long term (10 months) and document the progression of infection (n = 84) after BNT162b2 vaccination in a real environment, the humoral response was determined by ELISA immunoassay, and all the participants in the group showed IgG anti-S1 antibodies after vaccination. Anti-RBD IGG S1 antibodies were lost in almost half of the participants at 10 months. These data are consistent with those observed in our and other studies that studied serology up to 32 weeks after receiving mRNA vaccines [36,37,38]. Other studies have reported the clinical performance of the Abbott trial [31,35,39] with results similar to ours.

The decay of serum antibodies and neutralizing antibodies against SARS-CoV-2 has been extensively documented [40,41], although several studies have highlighted that vaccine-induced antibody levels also persist 6 months after the second dose [42].

Although there are many LFIA studies with CE markings on the market, three studies showed that the sensitivity and specificity values were similar to those of the EUROIMMUN trial [29,33,43]. However, another study evaluating six commercial assays for antibody detection at different levels of natural disease severity included three lateral flow tests (LFTs) (Acro IgM/IgG, CTK IgM/IgG, and Livzon IgM/IgG) and three ELISA assays (EUROIMMUN IgA and IgG tests and the Wantai IgM test) in 200 blood donors, and Livzon IgG LFT had the highest specificity (98.5%), followed by EUROIMMUN IgG ELISA (96.2%). The specificity of the EUROIMMUN IgA ELISA tests improved (97.5%), allowing us to conclude that the three evaluated LFIA tests are not suitable for diagnosis in mild cases of the disease and that ELISA trials are recommended in these settings. On the other hand, for the evaluation of seroprevalence, the IgG tests with high specificity, i.e., either ELIA or LFIA, could be appropriate [44]. There have been no studies describing the diagnostic performance of Livzon IgG in vaccinated subjects. Therefore, the results obtained in our study with sensitivity and specificity values comparable to the ELISA and CLIA tests as well as the agreement between the ELIA, CLIA, and LFIA tests of 99% (k = 0.964) at the second and third month after vaccination makes this test a very useful tool for monitoring vaccinated patients.

Another interesting finding was the homogeneity of the response of IgG antibodies in patients vaccinated against S and S1 proteins, as demonstrated by studies with other quantitative and qualitative immunoassays [29,45]. These results are consistent with those obtained in another study in which two automated immunological assays (the Abbott SARS-CoV-2 IgG CLIA and the EUROIMMUN anti-SARS-CoV-2 IgG/IgA ELISA) and a lateral flow immunoassay (the LFIA NG IgG-IgG COVID-19 test) were performed, with the results showing sensitivity for IgG detection of 100.0% in all trials. The overall specificity of IgG was higher for the CLIA and LFIA tests (more than 98%) than for the ELISA test (95.8%). The best agreement was observed between the CLIA and LFIA tests (97%; k = 0.936). Therefore, this study showed that the NG-Test lateral flow test is reliable and accurate for routine use in clinical practice for the detection of post-GG neutralizing Ig antibody vaccinations, as in the case with the Livzon test in our study. Another study in Australia with another lateral flow test, COVID-19 Nab-Test^TM^, using a cohort of vaccinated humans, correlated closely with data from the microneutralization trial (100% specificity and 96% sensitivity at a microneutralization limit of 1:40) [46].

Although quantitative reference tests are the gold standard and can identify IgG anti-SARS-CoV-2 antibodies against the binding domain of vaccine-generated S1 receptors, a strong correlation was observed between the levels of RBD-binding antibodies and the neutralizing antibodies of SARS-CoV-2 in patients, thus supporting the use of the RBD antigen in diagnostic serological tests. RBD-specific antibody levels corre-lated with the anticancer neutralization of SARS-CoV-2 [47]. With the Livzon LFIA IgG test against spike protein S, we showed a high correlation between our LFIA test and conventional neutralization, which is applicable in human blood and plasma samples. A quick test (15 min) that consists of pricking a finger and obtaining a few drops of whole blood, placing it in the well, adding the reagent and obtaining a qualitative evaluation by eye, and visually comparing the intensity of the test line with the reference line should provide sufficient information on whether an individual has an acceptable level of immunity, overcoming the limitations associated with current testing, including the need for a venous blood draw, laboratory equipment, and a slow change in results that lasts a minimum of three hours.

### 4.1. The Added Value of This Study

We evaluated the clinical performance of lateral flux, which can measure the levels of neutralizing antibodies with an RBD against protein S, and compared it with an ELISA test to measure antibodies against protein S1 and an ELISA test to quantitatively measure antibodies against proteins S1/S2. The results showed high sensitivity and specificity as well as a high degree of concordance and a high correlation with the gold-standard microneutralization test.

The performance of the LFIA test was established at different times after vaccination in a 6-month follow-up period in which several SARS-CoV-2 variants coexisted with homogeneous results, suggesting that this essay is easily adaptable to new emerging variants.

The Livzon test can be performed with whole blood samples from finger pricks or with plasma samples collected, thus providing rapid reading of the protection level based on the detection of antibodies at the point of care, in contrast to other evaluated tests that measure quantitative-type RBD antibodies that need more or less complex laboratories. It is thus possible to reach larger population groups when used carefully and evaluate collective immunity without saturating laboratory capacity.

These findings indicate that a fast, reliable, inexpensive, and reproducible test can be used in multiple scenarios in clinical practice, including supporting improved and individualized vaccination programs, especially in critical environments over time; obtaining the immunization status in high-risk people and health workers; or detecting donors of plasma therapy or convalescent antibodies. In cases of national or international mobility, a regular assessment of immune status may be required to apply recall doses while improving the use of health resources.

### 4.2. Limitations of the Study

As this study was limited to a trial of 300 subjects, it does not allow generalization in the general population. It also aimed to evaluate the seroconversion of previously healthy patients vaccinated in a given province, making it difficult to establish their noninferiority with other methods.

Another limitation of this study concerns the underlying premise that we will need to occasionally test people in order to assess their antibody status and risk of infection. While this is likely to be valuable, it should be recognized that even if/when antibody titers fall, memory B and T cells may provide rapid and effective protection after exposure.

Another important limitation is the fact that the study was conducted on healthy people without a documented history of infection and with negative RT-PCR and negative ELISA tests at the beginning of the study so as not to interfere with the natural immunity provided by infection. Therefore, comparisons could not be made between vaccinated people and those who have had natural infection based on their immunity.

Moreover, no comparisons could be made with standard viral neutralization tests, and cellular response analysis was not conducted due to limitations in the means and economics of our facilities.

## 5. Conclusions

Our study showed an equivalent clinical yield for IgG from three immunoassays (ELISA, CLIA, and LFIA) 21 days after vaccination. Unsurprisingly, all three trials had low sensitivity after the first dose at 21 days. It progressively increased from the second dose and reached the maximum during the second and third months after vaccination before slowly decreasing after the sixth month as neutralizing antibodies to SARS-CoV-2 decreased. The results with the LFIA test showed sensitivity and specificity values comparable to the ELISA and CLIA tests, and the agreement between the ELIA, CLIA, and LFIA tests was 99% (k = 0964) at the second and third month after vaccination. It was closely correlated with the data from the microneutralization panel, offering a positive agreement between 99% and 100% and a negative agreement between 88% and 100%. These results are consistent with those of other studies, suggesting that the post-antibody testing response to vaccination is an important and feasible tool to monitor people after vaccination and/or people who do not need more doses due to a previous infection of SARS-CoV-2. Therefore, serological tests may be useful to confirm the previous transmission of SARS-CoV-2, perform epidemiological seroprevalence studies, understand the seroconversion rates and duration of immunity conferred by vaccines, and help establish correct immunization guidelines.

## Figures and Tables

**Figure 1 jcm-11-07534-f001:**
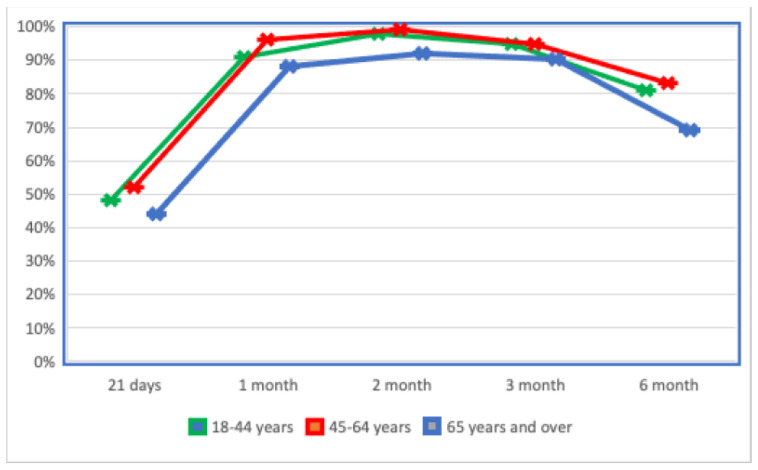
The proportion of seroconversions by age group. This figure shows the immune response against mRNA vaccines in the age groups, demonstrating how it is short-lasting in the group aged over 65 years of age.

**Table 1 jcm-11-07534-t001:** Sensitivity, specificity, predictive value, and likelihood ratio of serologic IgG assays.

ELISA Assay	21 Days	1 Month	2 Month	3 Month	6 Month
S % (IC 95%)	60.3 (42.7–87.6)	88.6 (70.5–87.1)	95.8 (89.7–98.4)	98.7 (91.5–99.7)	83.1 (75.3–89.7)
E % (IC 95%)	75.6 (66.9–82.7)	87.5 (83.891.9)	97.9 (92.1–96.6)	99.4 (96.1–99.9)	88.5 (85.3–99.6)
PPV % (IC 95%)	74.8 (65.3–83.5)	97.8 (92.8–99.7)	98.3 (92.1–99.6)	99.1 (90.9–99.9)	58.3 (42.7–66.8)
NPV % (IC 95%)	51.3 (43.8–58.8)	87.2 (69.3–56.4)	98.3 (92.3–99.6)	99.1 (96.6–99.9)	58.7 (42.8–78.7)
PLR (IC 95%)	2.06 (1.55–4.79)	7.1 (3.1–16–9)	28.3 (15.4–44.3)	60.1(21.2–205.5)	12.1 (6.2–25.6)
NLR (IC 95%)	0.6 (0.5–0.8)	0.3 (0.1–09)	0.2 (0.1–0.8)	0.1 (0.09–0.29)	0.05 (0.01–0.29)
CLIA assay	21 days	1 month	2 month	3 month	6 month
S % (IC 95%)	64.2 (46.4–92.5)	86.1 (74.5–91.1)	96.2 (87.6–97.5)	98.1 (99.9–99.5)	80.3 (73.2–88.6)
E % (IC 95%)	79.1 (66.6–85.8)	89.5 (85.9–93.9)	98.5 (93.2–97.7)	99.9 (97.3–100)	86.2 (85.3–99.6)
PPV % (IC 95%)	74.8 (65.3–83.5)	97.8 (92.8–99.7)	98.3 (92.1–99.6)	99.1 (90.9–99.9)	58.3 (40.5–64.7)
NPV % (IC 95%)	55.6 (45.8–60.8)	88.1 (70.4–57.5)	97.9 (91.2–99.8)	99.3 (96.3–99.9)	59.8 (41.8–77.7)
PLR (IC 95%)	3.3 (1.84–5.90)	8.3 (4.4–17.8)	30.2 (17.3–46.3)	72.3 (41.1–255.5)	18.1 (8.5–27.8)
NLR (IC 95%)	0.5 (0.3–0.9)	0.2 (0.1–0.6)	0.15 (0.1–0.9)	0.12 (0.8–0.29)	0.03 (0.01–0.27)
LFIA assay	21 days	1 month	2 month	3 month	6 month
S % (IC 95%)	60.6 (53.4–67–3)	80.2 (70.5–87.1)	94.8 (67.5–98.5)	97.8 (61.5–99.7)	77.6 (65.2–85.1)
E % (IC 95%)	70.1 (84.8–96.3)	86.7 (63.2–79.7)	97.9 (92.1–96.6)	98.4 (85.2–99.4)	83.1 (68.6–94.3)
PPV % (IC 95%)	75.9 (68.3–82.2)	96.1 (94.8–98.7)	98.1 (92.1–99.6)	97.1 (90.9–99.2)	48.5 (12.3–191)
NPV % (IC 95%)	53.3 (44.4–60.9)	87.9 (91.3–99.6)	98 (92.3–99.6)	96 (89.6–98.7)	48 (12.8–189)
PLR (IC 95%)	2.1 (1.52–2.70)	9.1 (8.56–10.62)	30.1 (12.8–96.2)	72.6 (43.1–96.2)	13.1 (6.4–26.9)
NLR (IC 95%)	0.56 (0.46–0.69)	0.3 (0.1–0.5)	0.2 (0.08–0.4)	0.1 (0.7–0.3)	0.05(0.0–0.3)

S: sensitivity; E: specificity; PPV: positive predictive value; NPV: negative predictive value; PLR: positive likelihood ratio; NLR: negative likelihood ratio; ELISA: EUROIMMUN anti-SARS-CoV-2 ELISA IgG S1 test; CLIA: anti-SARS-CoV-2 IgG S1/S2 LIAISON test; LFIA: Livzon anti-SARS-CoV-2 IgG S test. The table shows the evolution of the diagnostic performance of the three immunoassays over time, demonstrating how it increases in all parameters from 21 days to 3 months and begins to decrease after 6 months.

**Table 2 jcm-11-07534-t002:** The concordance of IgG serologic tests.

Concordance	Assays	ELISA% (Kappa)	CLIA% (Kappa)
General	CLIA	97% (0.938)	
	LFIA	96% (0.919)	97% (0.941)
21 days	CLIA	88% (0.666)	
	LFIA	95% (0.914)	91% (0.753)
1 month	CLIA	96% (0.931)	
	LFIA	96% (0.919)	98% (0.954)
2 month	CLIA	100% (1.00)	
	LFIA	99% (0.964)	99% (0.964)
3 month	CLIA	100% (1.00)	
	LFIA	99% (0.964)	99% (0.964)
6 month	CLIA	87% (0.625)	
	LFIA	93% (0.845)	91% (0.812)

ELISA: EUROIMMUN anti-SARS-CoV-2 IgG S1 assay; CLIA: anti-SARS-CoV-2 IgG S1/S2 LIAISON assay; LFIA: Livzon anti-SARS-CoV-2 IgG S assay. The degree of agreement of the LFIA test was similar to those of the ELISA and the CLIA tests throughout the follow-up.

**Table 3 jcm-11-07534-t003:** The results of the seroconversion panel.

Panel Member	Day	ELISA ^a^	CLIA ^b^	LFIA ^c^
1	1	0.1	5.4	−
2	21	1.2	15	−/+
3	30	7.16	39.10	+
4	60	12.14	45.60	+
5	90	10.53	40.90	+
6	120	5.20	27.70	+
7	150	4.80	19.57	+
8	180	3.40	17.30	+

^a^ ELISA (EUROIMMUN anti-SARS-CoV-2 IgG S1 assay): units defined by the manufacturer (AU/mL): <0.8—negative; 0.8 to <1.0—limit; 1.1—positive. ^b^ CLIA (LIAISON de Diasorin anti-SARS-CoV-2 IgG S1/S2 assay): units defined by the manufacturer (AU/mL): <12.0—negative; 15.0—limit; >15—positive. ^c^ LFIA (Livzon anti-SARS-CoV-2 IgG S test): red-dyed band—+; nondyed band—−; faint staining—−/+, indicating the limit values and thus a very low concentration of antibodies.

## Data Availability

Not applicable.

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
