# Peer review of "An Evaluation of Serological Tests to Determine Postvaccinal Immunity to SARS-CoV-2 by mRNA Vaccines"

_jcm, 2022, doi:10.3390/jcm11247534_

Round 1

Reviewer 1 Report

This article relevant in current context, but my observations are given bellow.

Introduction

1.       This para related to methods of diagnosis SARS- CoV-2 not desired.

Early detection of all cases compatible with COVID-19 is one of the key points to control transmission. The performance of Diagnostic Tests for Active Infection by SARS- CoV-2 (PDIA) should be aimed mainly at the early detection of cases with transmission capacity, prioritizing this use over other strategies. Currently, there are two active infection detection tests available, a rapid resistance detection test (Antigen Rapid diagnostic test, Ag-RDT)3 and viral RNA detection by RT-PCR or an equivalent molecular technique Detection of viral RNA by real-time reverse transcriptase-polymerase chain reaction  (RT-PCR) in respiratory tract samples is considered the standard method for detection and diagnosis in the early phase of infection6, 7 .

2.       although it is unknown how long they protect vaccinated subjects17, so there is concern about the efficacy and durabil ity of the vaccine protection.

But, studies are there related to protective duration of various vaccines and declining trend of antibody titre. It is better to narrate those studies and provide explanation how this study will give better information.

Methods

3.        prospective study included 300 healthy subjects, Explain how you reached with this sample size of 100 each.

Results

4.       positive predictive value (PVP) and negative predictive value (PVN), positive probability ratio (CPP) and negative probability ratio (CPN), are differently abbreviated in 1st table, Have a look.

Discussion

5.       You have to elaborate about other competitors related to your methods of detecting vaccine induced anti-SARS- CoV-2 IgG antibodies. With sample size of 300 only, how establish your non-inferiority?

Conclusion

6.       This study unable to provide information regarding booster dose of SARS CoV-2 vaccination.

Author Response

Dear Reviewer:

We are very grateful for your kind and accurate comments, as well as for your valuable suggestions that allow us to make a higher quality article. We have updated the manuscript jcm-2021419 as follows:

  1. Introduction:

Taking into account your suggestions we have rewritten the introduction and explained the purpose of the study in a more understandable way.

  1. Methods:

Subjects were recruited through systematic sampling with random start of the SACYL database of 115,075 subjects candidates to receive mRNA vaccines (9.6% with modern) in the province of León until the completion of 300 subjects at the time of vaccination, distributed in 3 age groups: 100 subjects over 18 years and under 45 years; 100 subjects between 45 - 65 years and another 100 over 65 years to avoid possible age-induced variability.

  1. Results

Corrected positive predictive value (PVP) and negative predictive value (PVN), positive probability coefficient (CPP) and negative probability coefficient (CPN) in the first table and included the result of a seroconversion panel (Table 3)  to compare our LIVZON LFIA test with the gold standard microneutralisation test on a seroconversion panel to compare positive and negative match grades.

  1. Discussion:

The discussion has been rewritten and enriched, following their suggestions, giving more details about other competitors with our methods of detecting anti-SARS-CoV-2 IgG antibodies induced by vaccines in different environments. In addition, the study’s strengths have been highlighted in the section (added value of this study) and its weaknesses (limitations section).

  1. Conclusions:

It has been written again making it more comprehensible and in line with the objective of the study and the results obtained.

As you correctly point out this study cannot provide information on the booster dose of SARS-vaccinationCoV-2, although it can be intuited by longitudinal monitoring and determination of the response generated by vaccines with the three immunoassays, The aim of this study was to evaluate the clinical performance of the LFIA immunoassay compared with ELIA and CLIA immunoassays available in Europe for the detection of Ig G antibodies generated by mRNA vaccines against SARS-CoV-2. The main result together is that the device LFIA Livzon IgG anti S, provides a quick result on the presence of antibodies at the point of care to people who have received vaccines with a sensitivity, specificity, degree of agreement and correlation when compared with the gold standard comparable to that of ELISA and CLIA.

  1. 6. References:

References supporting obsolete information have been withdrawn and new references have been introduced more in line with the current situation and the state of knowledge of the subject.

In the hope that all the changes introduced will conform to the indications you have given us, we would be very grateful to receive your signature on the review report.

Sincerely

Ángel Díaz Rodríguez, MD, PhD, MsCs

19/11/2022

Reviewer 2 Report

Assessing the effectiveness of different vaccines against SARS-CoV-2 is essential for the development of public health policies. However, the authors had access to rare and important biological samples, follow-up before vaccination and at different points after vaccination, and only assessed IgG levels in these samples, comparing three commercial assays already validated and published in the literature. The material could have been better explored with standard viral neutralization assays, as well as cellular response analysis. Moreover, the results are basic and little explored and discussed. 

Author Response

Dear Reviewer:

We are very grateful for your kind and accurate comments, as well as for your valuable suggestions that allow us to make a higher quality article. We have updated the manuscript jcm-2021419 as follows:

  1. Introduction:

Taking into account your suggestions we have written again the introduction and have explained the purpose of the study in a more understandable way, as well as removing references irrelevant to the study and adding a large number of very relevant references.

  1. Methods:

The methodology has been better explained in different aspects such as patient recruitment and sampling. Healthy patients were chosen, with no history of SARS-Cov-2 infection, with negative RT-PCR performed at the time of inclusion before vaccination and negative ELISA before vaccination because we wanted to obtain immunoassay performance, Especially the experimental LFIA Livzon, in vaccinated patients, for which this test is not validated in the literature.  These three tests have been validated in the literature for the detection of antibodies in subjects who have suffered natural infection, as you point out very well in your report, but the lateral flow test has not been validated to determine postvaccine IgG antibodies in any study of which this working group is currently aware.

We evaluated the clinical performance of the three immunoassays only in patients from the start of vaccination and followed them for 6 months. The experimental method to perform the clinical evaluation of the test was performed by calculating sensitivity, specificity, positive predictive value (PVP) and negative predictive value (PVN), positive probability ratio (CPP) and negative probability ratio (CPN), as well as the correlation by Cohen’s Kappa index for each serological test. If the degree of concordance meets the above clinical requirements, both methods are considered equivalent. To verify the accuracy and applicability of the test in clinical practice, the LIFIA immunoassay must have a sensitivity, specificity and degree of concordance greater than 90%.

To compare our LIVZON LFIA test with the gold standard microneutralization test on a seroconversion panel to compare positive and negative concordance degrees.

Given the technological and economic limitations of the laboratories of our university and our reference hospitals, it was not possible to carry out the tests of cellular response, microneutralization and/or standard viral neutralization, obtaining the correlation from a seroconversion panel.

  1. Results

To complete the results, the result of a seroconversion panel was included in section 3.5. of the results and a new table (Table 3)  to compare our LIVZON LFIA test with the gold standard microneutralisation test on a seroconversion panel to compare the positive and negative concordance degrees, with the results obtained with ELISA and CLIA.

  1. Discussion:

The discussion has been rewritten and enriched, giving more details about other competitors with our methods of detecting anti-SARS-CoV-2 IgG antibodies induced by vaccines in different environments. In addition, the study’s strengths have been highlighted in the section (added value of this study) and its weaknesses (limitations section).

  1. Conclusions:

It has been written again making it more comprehensible and in line with the objective of the study and the results obtained.

The aim of this study was to evaluate the clinical performance of the LFIA immunoassay compared with ELIA and CLIA immunoassays available in Europe for the detection of Ig G antibodies generated by mRNA vaccines against SARS-CoV-2. The main result together is that the device LFIA Livzon IgG anti S, provides a quick result on the presence of antibodies at the point of care to people who have received vaccines with a sensitivity, specificity, degree of agreement and correlation when compared with the gold standard comparable to that of ELISA and CLIA.

  1. References:

References supporting obsolete information have been withdrawn and new references have been introduced more in line with the current situation and the state of knowledge of the subject.

Hoping that all the changes introduced will conform to the suggestions you have given us, we would be very grateful to receive your signature on the review report.

Sincerely

Angel Díaz Rodríguez, MD, PhD, MsCs

19/11/2022

Reviewer 3 Report

The manuscript by Calvo et al. scientifically weak manuscript. The manuscript fails to provide anything new or at least interesting. I have following comments on the manuscript.

1). Line 49-50. The sentence is not clear.

2). Line 51-53. It is very well studied. Please look the published work carried  by the Nussenzweig laboratory at the Rockefeller University.

3). Line 57-58. Not true. This is well studied and is the reason why booster doses are given.

4). Line 61-65. Absolutely wrong.

Author Response

Dear Reviewer:

We are very grateful for your kind and accurate comments, as well as for your valuable suggestions that allow us to make a higher quality article. We have updated the manuscript jcm-2021419 as follows:

  1. Introduction:

Taking into account your suggestions we have written again the introduction and have explained the purpose of the study in a more understandable way, as well as removing references irrelevant to the study and adding a large number of very relevant references.

  1. Methods:

The methodology has been better explained in different aspects such as patient recruitment and sampling. Healthy patients were chosen, with no history of SARS-Cov-2 infection, with negative RT-PCR performed at the time of inclusion before vaccination and negative ELISA before vaccination because we wanted to obtain immunoassay performance, Especially the experimental LFIA Livzon, in vaccinated patients, for which this test is not validated in the literature.  These three tests have been validated in the literature for the detection of antibodies in subjects who have suffered natural infection, as you point out very well in your report, but the lateral flow test has not been validated to determine postvaccine IgG antibodies in any study of which this working group is currently aware.

We evaluated the clinical performance of the three immunoassays only in patients from the start of vaccination and followed them for 6 months. The experimental method to perform the clinical evaluation of the test was performed by calculating sensitivity, specificity, positive predictive value (PVP) and negative predictive value (PVN), positive probability ratio (CPP) and negative probability ratio (CPN), as well as the correlation by Cohen’s Kappa index for each serological test. If the degree of concordance meets the above clinical requirements, both methods are considered equivalent. To verify the accuracy and applicability of the test in clinical practice, the LIFIA immunoassay must have a sensitivity, specificity and degree of concordance greater than 90%.

To compare our LIVZON LFIA test with the gold standard microneutralization test on a seroconversion panel to compare positive and negative concordance degrees.

Given the technological and economic limitations of the laboratories of our university and our reference hospitals, it was not possible to carry out the tests of cellular response, microneutralization and/or standard viral neutralization, obtaining the correlation from a seroconversion panel.

  1. Results

To complete the results, the result of a seroconversion panel was included in section 3.5. of the results and a new table (Table 3)  to compare our LIVZON LFIA test with the gold standard microneutralisation test on a seroconversion panel to compare the positive and negative concordance degrees, with the results obtained with ELISA and CLIA.

  1. Discussion:

The discussion has been rewritten and enriched, giving more details about other competitors with our methods of detecting anti-SARS-CoV-2 IgG antibodies induced by vaccines in different environments. In addition, the study’s strengths have been highlighted in the section (added value of this study) and its weaknesses (limitations section).

  1. Conclusions:

It has been written again making it more comprehensible and in line with the objective of the study and the results obtained.

The aim of this study was to evaluate the clinical performance of the LFIA immunoassay compared with ELIA and CLIA immunoassays available in Europe for the detection of Ig G antibodies generated by mRNA vaccines against SARS-CoV-2. The main result together is that the device LFIA Livzon IgG anti S, provides a quick result on the presence of antibodies at the point of care to people who have received vaccines with a sensitivity, specificity, degree of agreement and correlation when compared with the gold standard comparable to that of ELISA and CLIA.

  1. References:

References supporting obsolete information have been withdrawn and new references have been introduced more in line with the current situation and the state of knowledge of the subject.

Additionally, it has undergone extensive revisions in English, using mdpi’s editing services: https://www.mdpi.com/authors/english

Hoping that all the changes introduced will conform to the suggestions you have given us, we would be very grateful to receive your signature on the review report.

Sincerely

Angel Díaz Rodríguez, MD, PhD, MsCs

19/11/2022

Round 2

Reviewer 1 Report

Well revised.

Reviewer 3 Report

The manuscript can be accepted now.